# Effect of Strength Training Protocol on Bone Mineral Density for Postmenopausal Women with Osteopenia/Osteoporosis Assessed by Dual-Energy X-ray Absorptiometry (DEXA)

**DOI:** 10.3390/s22051904

**Published:** 2022-02-28

**Authors:** Iulian Ștefan Holubiac, Florin Valentin Leuciuc, Daniela Maria Crăciun, Tatiana Dobrescu

**Affiliations:** 1Department of Physical Education and Sport, Stefan cel Mare University, 720229 Suceava, Romania; holubiac.iulianstefan@usm.ro (I.Ș.H.); daniela.craciun@usm.ro (D.M.C.); 2Department of Physical Education and Sport Performance, Vasile Alecsandri University, 600115 Bacau, Romania; tatiana.dobrescu@ub.ro

**Keywords:** osteoporosis, strength training, osteopenia, bone mass, DEXA

## Abstract

This study aims to introduce a resistance training protocol (6 repetitions × 70% of 1 maximum repetition (1RM), followed by 6 repetitions × 50% of 1RM within the same set) specifically designed for postmenopausal women with osteopenia/osteoporosis and monitor the effect of the protocol on bone mineral density (BMD) in the lumbar spine, assessed by dual-energy X-ray absorptiometry (DEXA). The subjects included in the study were 29 postmenopausal women (56.5 ± 2.8 years) with osteopenia or osteoporosis; they were separated into two groups: the experimental group (*n* = 15), in which the subjects participated in the strength training protocol for a period of 6 months; and the control group (*n* = 14), in which the subjects did not take part in any physical activity. BMD in the lumbar spine was measured by DEXA. The measurements were performed at the beginning and end of the study. A statistically significant increase (Δ% = 1.82%) in BMD was observed at the end of the study for the exercise group (0.778 ± 0.042 at baseline vs. 0.792 ± 0.046 after 6 months, *p* = 0.018, 95% CI [−0.025, −0.003]); while an increase was observed for the control group (Δ% = 0.14%), the difference was not statistically significant (0.762 ± 0.057 at baseline vs. 0.763 ± 0.059, *p* = 0.85, 95% CI [−0.013, 0.011]). In conclusion, our strength training protocol seems to be effective in increasing BMD among women with osteopenia/osteoporosis and represents an affordable strategy for preventing future bone loss.

## 1. Introduction

The introduction of the bone mass assessment method using dual-energy X-ray absorptiometry (DEXA) measurement is an important step for clinical trials aimed at assessing bone density. DEXA has certain advantages, such as increased accuracy and the low radiation dose to which the subject is exposed. DEXA investigation can measure bone mass for the spine, hip, and forearm, thereby helping in the diagnosis of osteopenia or osteoporosis. DEXA is a non-invasive diagnostic technique used to determine bone density and it is a reference method in this domain [1].

The combination of bone density and bone quality (obtained from bone mass assessment) mainly reflects bone strength. When talking about bone quality, several variables are considered, such as structural changes, the relationship between osteolysis and osteogenesis, collagen structure, possible bone damage (e.g., fractures or microfractures), and the level of bone mineralisation. Osteoporosis can be the result of lack of calcium, as well as other minerals in bones, and these deficiencies can make bone weaker, more fragile, and more prone to injuries and fractures, even following minor trauma [2,3]. Osteoporosis can be classified according to the causative factor; however, in the case of primary osteoporosis, it affects about 80% of women and about 60% of men. Primary osteoporosis includes idiopathic osteoporosis (i.e., the cause of osteoporosis is unknown), Type I osteoporosis (which is caused by a lack of oestrogen, a common condition in postmenopausal women), and Type II (degenerative) osteoporosis. Osteoporosis caused by oestrogen deficiency is characterised by loss of bone density, which may be accompanied by fractures in the femoral neck, lumbar spine, and distal radius. Oestrogen plays a protective role in the bone system, and when women enter the postmenopausal period and oestrogen levels drop significantly, osteolysis becomes more pronounced compared with osteogenesis because the lack of oestrogen tends to cause an imbalance between osteogenesis and osteolysis (in favour of osteolysis), making bones more fragile. In other words, osteoclasts (the bone cells responsible for osteolysis) become more active than osteoblasts (the bone cells responsible for osteogenesis). Loss of cortical and trabecular bone, as well as proximal fractures of the humerus and tibia, femoral neck, and pelvis, are specific to Type II osteoporosis. Oestrogen plays a critical role in regulating several factors that are responsible for osteogenesis, such as RANKL (nuclear activator receptor kappa-B ligand factor) in osteoblasts, IL-1, IL-6, TNF-α, and M-CSF. Oestrogen also stimulates osteoprotegerin activity in osteoblasts, an action that results in the apoptosis of osteoclasts and prevents the apoptosis of osteocytes (bone cells). When oestrogen levels decrease (as in the case of menopause), the rate of osteolysis increases and the rate of osteogenesis decreases, which leads to a decrease in bone mass. To improve mobility, bone strength, and physical function, and to prevent fractures (as a consequence of falls), resistance training should be practiced along with balance exercises and weight-bearing activities [4]. Resistance exercises seem to be effective even amongst older women when it comes to bone mass [5], with recommended intensities between 70% and 80% of one maximum repetition (1RM), performed two to three times a week being an affordable and efficient solution for increasing bone mass amongst postmenopausal women [6,7].

This study aims to evaluate the influence of a resistance training protocol for lumbar spine bone mineral density (BMD) amongst women with postmenopausal osteopenia/osteoporosis assessed by DEXA.

## 2. Materials and Methods

Twenty-nine non-smoking women with postmenopausal osteopenia/osteoporosis, a body mass index (BMI) of ≤25, and no physical exercise contraindications were included in the study. The subjects had not participated in any exercise program in the last 3 months, and all of them were sedentary people (less than 60 min of exercise per week). BMD was measured using DEXA analysis (Hologic Horizon, Santa Clara, CA, USA) and radiological examination (see Figure 1 and Figure 2). Skeletal BMD can be measured using DEXA, which is considered by some authors to be one of the most effective methods of diagnosing osteoporosis or osteopenia [8]. Currently, DEXA is the standard reference for diagnosing osteoporosis [9,10]. It is a precise evaluation method that involves exposing the areas to be evaluated to a small amount of X-rays and allows the areas of interest to be objectively measured. In this method, the two X-rays are absorbed differently in the bone, and the BMD is calculated in g/cm^2^ using simultaneous equations. Among the results offered by DEXA investigation is some information on bone mineral content (BMC) given in g, area measured in cm^3^, and BMD given in g/cm^2^ [11]. The measurement was compared with two reference values: one for young adults (30 years, which gives a T-score) and one for people of the same age as the evaluated person (which gives a Z score) [12]. DEXA has certain strengths, such as not exposing the patient to a high dose of radiation (1–6 μSv) and having a short time scan (1–2 min). Following the DEXA investigation, the software recorded the values obtained and displayed them on the screen. The T-score obtained by the subject being examined refers to her bone mass, taking as reference an individual of the same gender with peak bone mass. A classification was established depending on the score obtained: normal bone mass density (score between −1 and 0 or higher), osteopenia (between −1.1 and −2.4), and osteoporosis (a score of −2.5 or less). The Z score obtained refers to the BMD of the scanned subject compared with a subject of the same age and weight (see Figure 2).

Medical investigations (DEXA and radiographs) were conducted by a technician and the radiographs was interpreted by a specialist. The T-score provided information on whether the subject could be included in the study (the condition being that the subjects had osteopenia or osteoporosis), and the lumbar spine BMD was the quantitative information we used to make pre- and post-test comparisons. Table 1 and Table 2 and Figure 2 present information on BMD in each vertebra, but the last and the most important result is the total BMD, which is the reference result. With the help of radiography, we used visual information about the body and height of the vertebrae, the presence or absence of vertebral fractures, and the height of the intervertebral spaces. We could also observe if the subject had a history of other pathologies that would be a contraindication to performing the training protocol. Different from the DEXA investigation, which was made at the beginning and at the end of the study, the X-ray was used only at the beginning of the study to rule out the possibility of other pathologies in the lumbar area that could prevent the subjects from participating in the program.

The study involved women with osteopenia/postmenopausal osteoporosis who were separated into two groups, as shown in Figure 3. The exercise group was formed by volunteer women; the control group was formed by women who did not want to participate in the training protocol. All the subjects received the same treatment (0.5 µg of alfacalcidol daily). The subjects gave their written consent to the use of the data obtained and the study was conducted according to the guidelines of the Declaration of Helsinki. The flowchart of patient registration is shown in Figure 3.

The training protocol was designed by two physical therapists and two sports training specialists. The training program lasted 6 months; the duration of each training session was about 60 min and took place in the gym of Stefan cel Mare University of Suceava. Exercises included in the training protocol (seated hip abduction, seated machine dip, seated back extension, standing hip flexion, standing hip extension, seated hip adduction, horizontal leg press, prone hamstring curls, seated knee extension, and bicep curls) were performed on the machines. Only bodyweight squats were not performed on the machine. The first six exercises were performed in the first training session of the week and the other five exercises were performed in the second training session of the week. All exercises were performed in two sets of 12 repetitions (6 repetitions with an intensity of 70% of 1RM, followed by 6 repetitions with an intensity of 50% of 1RM). The only exception was bodyweight squats, where no extra weights were used to avoid overloading the lower limb joints. The pause between sets was 90 s. The subjects had two weeks to familiarise themselves with the exercises and learn the correct execution technique. Three days before the start of the training protocol, the subjects were tested for 1RM for each exercise, such that we could determine the 70% and 50% of 1RM. The 1RM represents the weight with which the subject can perform only one repetition. This test was repeated every 4 weeks, such that the future estimates of 70% and 50% could be made according to the new maximum repetition (1RM). The strength exercises were performed with an intensity of 40% of 1RM in the first two weeks with 12–15 repetitions per series, followed by 50% of 1RM in the third week. Starting in the fourth week, we switched to a method involving six repetitions with an intensity of 50% of 1RM followed by six repetitions with an intensity of 70% of 1RM within the same set. Given that fractures of the wrist, vertebral body, and femoral neck are the fractures that warn of the presence of osteoporosis, the exercises proposed in this protocol focused muscle groups that are involved in the movement of wrist, spine, and hip joints. Seated hip abduction targets the gluteus medius and gluteus minimus muscles. Standing hip flexion targets the iliopsoas and quadriceps femoris muscles (more precisely, rectus femoris muscles). The psoas major muscle originates in the adjacent margins of the bodies of the vertebrae (T_12_–L_5_) and intervening intervertebral disc, such that its contraction acts on the vertebral bodies. Standing hip extension is an exercise that strengthens the gluteus maximus muscle. Seated hip adduction mainly targets the adductor magnus, adductor longus, and adductor brevis muscles. Prone hamstring curls target the biceps femoris, semitendinosus, and semimembranosus muscles. Horizontal leg press targets the gluteus maximus, quadriceps femoris, and triceps surae muscles. Seated knee extension involves an isolated movement of the knee joint with excellent effect on the quadriceps femoris muscle. Seated machine dips stimulate the muscles of the upper limbs, shoulder, and thorax (pectoralis major, anterior deltoid, and triceps brachii). Bicep curls strengthen the muscles of the front of the arm and forearm (especially the biceps brachii, brachialis, and brachioradialis). Seated back extension exercises strengthen the back muscles. The muscles that are involved in this movement are the erectores spinae (iliocostalis, longissimus, and spinalis), aided by the quadratus lumborum and latissimus dorsi. With age, especially in women with osteopenia/osteoporosis, thoracic curvature (kyphosis) increases due to vertebral deformities. Thus, we selected the above exercise because it strengthens the back extensors to maintain a correct posture.

SPSS version 26 was used for statistical analysis. To determine the statistical significance, a paired sample *t* test was applied for within-group comparisons and an independent sample *t* test was applied for between-group comparisons. The Mann–Whitney U test was applied to the variable ‘age’ to determine if there existed significant differences between the two groups. A *p* value < 0.05 was considered statistically significant. The equation used for the paired sample *t* test is
(1)$$t=\frac{\bar{X_{D}}}{S_{D}/\sqrt{n}}$$
$\bar{X_{D}}$ = mean difference between the two groups (control and exercise groups for each variable: BMD_L1_, BMD_L2_, BMD_L3_, BMD_L4_ and BMD_total_); $S_{D}$ = standard deviation of the difference scores; and *n* = number of subjects. The equation for $S_{D}$ calculation is
(2)$$S_{D}=\sqrt{\frac{\sum_{x}2-(\sum_{x})2/n\;}{n-1}}$$

The equation used for the independent sample *t* test is
(3)$$t_{test}=\frac{M_{x\;\;}–\;M_{y}}{\sqrt{\frac{(\sum X^{2}-{\left(\sum X\right)}^{2}/N_{x})+(\sum Y^{2}-{\left(\sum Y\right)}^{2}/N_{y})*\left(\frac{1}{N_{x}}+\frac{1}{N_{xy}}\right)\;\;\;}{N_{x}+N_{y}-2}}}$$

$M_{x}$ = mean score for exercise group; $M_{y}$ = mean score for control group; $\sum X^{2}\;$ = sum of the squared *X* (control group) scores; ${\left(\sum X\right)}^{2}\;$ = sum of the *X* (control group) scores squared; $\sum Y^{2}\;$ = sum of the squared *Y* (exercise group) scores; ${\left(\sum Y\right)}^{2}\;$ = sum of the *Y* (exercise group) scores squared; $N_{x}$ = number of control group subjects; and $N_{y}$ = number of exercise group subjects.

## 3. Results

The baseline characteristics of the participants are shown in Table 1. For the exercise group, the weight of the subjects showed a significant decrease (Δ% = −1.93%) at the end of the study (64.5 ± 7.9 vs. 65.8 ± 7.4, t(14) = 3.11, *p* = 0.008, 95% CI [0.39, 2.14]). Within the control group, the weight of the subjects increased (Δ% = 1.13%) after 6 months (M = 63.9, SD = 7.6) compared with the initial results (M = 63.2, SD = 7.5, t(13) = −2.02, *p* = 0.065, 95% CI [0.05, −2.02]). Although a difference existed between the two groups, the statistical analysis showed that it was not statistically significant (t(27) = 0.21, *p* = 0.84, 95% CI [−5.31, 6.52]). The height of the subjects who participated in the weight training program decreased (Δ% = −0.12) after 6 months (160.8 ± 6.3 vs. 161.0 ± 6.3, t(14) = 1.87, *p* = 0.082, 95% CI [−0.03, 0.43]). No significant differences (Δ% = −0.09) were observed at the end of the study in the control group (M = 157.5, SD = 4.6) compared with the initial test (M = 157.6, SD = 4.7, t(13) = 1.47, *p* = 0.17, 95% CI [−0.07, 0.35]). The intergroup differences were not statistically significant (t(27) = 1.60, *p* = 0.12, 95% CI [−0.93, 7.53]).

BMI decreased (Δ% = −1.79) within the exercise group at the end of the study (24.9 ± 2.4 vs. 25.4 ± 2.6, t(14) = 2.71, *p* = 0.017, 95% CI [0.10, 0.81]); within the control group, there was an increase by 1.27% at the end of the study (M = 25.7, SD = 2.1) compared with the baseline (M = 25.4, SD = 2.1, t(13) = −2.04, *p* = 0.062, 95% CI [−0.66, 0.02]), but the differences were insignificant at the end of the study between the two groups (t(27) = −0.90, *p* = 0.37, 95% CI [−2.48, 0.96]; see Figure 4 and Table 2).

BMD showed an increase (Δ% = 3.01%) at the L_1_ lumbar spine after 6 months (M = 0.777, SD = 0.055) compared with the baseline (M = 0.754, SD = 0.057), but the difference was not statistically significant (t(14) = −1.90, *p* = 0.079, 95% CI [−0.048, 0.003]). For the same area, the subjects who did not participate in the weight training program showed a decrease (Δ% = −1.01%) after 6 months compared with the initial results (0.743 ± 0.061 vs. 0.750 ± 0.072, t(13) = 1.17, *p* = 0.26, 95% CI [−0.006, 0.022]). Although there existed a difference between the groups, the difference was not statistically significant (t(27) = 1.59, *p* = 0.12, 95% CI [−0.010, 0.078]).

At the L_2_ level, BMD showed an increase (Δ% = 1.44%) within the exercise group after 6 months (0.791 ± 0.061 vs. 0.780 ± 0.050), but the increase was not statistically significant (t(14) = −1.24, *p* = 0.24, 95% CI [−0.031, 0.008]). Within the control group, an increase (Δ% = 1.70%) was also observed at the end of the research compared with the baseline (t(13) = −1.74, *p* = 0.11, 95% CI [−0.029, 0.003]). Between the two groups, the difference was not statistically significant at the end of the study (t(27) = 0.61, *p* = 0.55, 95% CI [−0.034, 0.063]). At the end of the study, the exercise group showed an increase (Δ% = 2.55%) at the L_3_ level, but the increase was not statistically significant (0.805 ± 0.057 vs. 0.785 ± 0.054, t(14) = −2.11, *p* = 0.053, 95% CI [−0.040, 0.000]). At the same level, the control group showed a decrease in bone mass at the end of the study (Δ% = −1.29%) compared with the initial results (0.776 ± 0.066 vs. 0.786 ± 0.064, t(13) = 0.91, *p* = 0.38, 95% CI [−0.014, 0.034]). However, the difference was insignificant between the two groups (t(27) = 1.28, *p* = 0.21, 95% CI [−0.018, 0.076]). Both groups registered an extremely close increase in value at the level of the L_4_ lumbar spine (exercise group: Δ% = 0.68%, t(14) = −0.47, *p* = 0.65 and 95% CI [−0.030, 0.019]; control group: Δ% = 0.69%, t(13) = −0.51, *p* = 0.62, 95% CI [−0.027, 0.017]), but the difference not being statistically significant between the two groups (t(27) = 1.63, *p* = 0.12, 95% CI [−0.010, 0.089]). Total lumbar spine BMD showed a statistically significantly increase (Δ% = 1.82%) in the exercise group after 6 months (0.792 ± 0.046 vs. 0.778 ± 0.042, t(14) = −2.68, *p* = 0.018, 95% CI [−0.025, −0.003]). The control group also showed an improvement (Δ% = 0.14%) after 6 months (M = 0.763, SD = 0.059) compared with the initial results (M = 0.762, SD = 0.057), but the difference was not statistically significant (t(13) = −0.20, *p* = 0.85, 95% CI [−0.013, 0.011]). Although the experimental group showed a higher increase compared with the control group, the difference was not statistically significant at the end of the study (t(27) = 1.49, *p* = 0.15, 95% CI [−0.011, 0.069]).

## 4. Discussion

Radiological evaluation is still used in the diagnosis of osteoporosis, although it records at a low sensitivity. The changes that we can observe on the radiograph are represented by osteopenia, fractures, and the consequences of these deformities. Thus, a decrease of the vertebrae height is a consequence of the fractures, a particularly important aspect in the diagnosis of osteoporosis. In the case of radiographic investigations, for a person’s bones to show osteopenic signs, they have to lose about 20–30% of their bone mass, which is a fairly high percentage [13]. Therefore, this condition is a limitation of radiographic investigations. Some features of the osteoporotic bone, visible on radiographic investigations, are represented by the thinning of the cortical bone and deformities of the spine vertebrae [14,15]. However, one study found that hand radiographs are not as accurate compared with DEXA (dual X-ray) measurements [16]. The diagnosis of osteopenia or osteoporosis using radiographic imaging is not highly accurate because it is influenced by the position of the patient’s body during the investigation. The diagnosis depends on the level of training and experience of the doctor who interprets the X-ray. Identifying people at increased risk of a fracture within a year or two of the assessment data (imminent risk of fracture) is a new concept that can be useful in selecting people to receive immediate treatment and a program of fall prevention [17].

DEXA investigation is used to measure the BMC of the human skeleton or certain areas considered vulnerable to fracture (hip, spine, and forearm). DEXA can help determine the strength and strength of bone, which are influenced by bone size. The best anatomical area to measure BMD is at the hip joint (hip).

The BMD resulting from this investigation is closely related to the thickness of the bone trabeculae, but also to their number. Cortical bone thickness and bone section area are also related to BMD, these parameters being in turn subject to hormonal influences, lifestyle, physical activity, hereditary antecedents, and constituting factors that interfere with the evaluation and interpretation of DEXA. The areas of interest used in this technique are at the hip joint and at the lumbar spine (L_1_–L_4_). Investigation of the hip joint is preferred in patients, especially after the age of 65 when other degenerative diseases progress and may interfere with the bone mass present in the spine. Other areas of interest are represented by the middle and distal radius. According to the National Osteoporosis Foundation, the indications for DEXA investigation are: (a) women aged 65 and over and men aged 70 and over; (b) postmenopausal women and men ≥50 years of age with other risk factors involved; (c) postmenopausal women and men ≥50 years of age who have suffered a fracture at or after the age of 50; and (d) adults who suffer from rheumatoid arthritis or who use medications, such as glucocorticoids, and who have low bone mass or bone loss [18,19]. This routine investigation is not recommended for premenopausal woman. DEXA has some limitations, although it is a quick and an inexpensive method of diagnosing osteoporosis. This technology cannot produce 3D images and cannot distinguish between cortical and trabecular bones [20]. During the DEXA investigation, the operator may be exposed to a low dose of radiation [20,21,22].

The effect of resistance training on BMD depends on the duration of the program, the intensities used, and the treatment followed.

A study conducted in 2013 on 21 women with osteopenia/osteoporosis reported positive results for the experimental group (61.9 ± 5.0 years; *n* = 10) compared with the group control (66.7 ± 7.4 years; *n* = 11) at the end of 12 weeks of weight training, with the program performed three times a week. The program started with two warm-up series of 8–12 repetitions with an intensity of 50% of 1RM, followed by four series of 3–5 repetitions with an intensity of 85–90% of 1RM, with the rest between the series being 2–3 min. At the level of the experimental group, BMC registered a significant increase in the lumbar spine (Δ% = 2.9, *p* = 0.012) [23]. The intensities used in this study were extremely high (85–90%); in our case, we used lower intensities to avoid the risk of injury. Given that overloading the joints by using extremely heavy weights can increase the risk of bone cartilage damage, our decision to use lower intensities is understandable. Although the improvement observed in that study is more obvious, with an increase 1% higher compared with our work, the subjects underwent a treatment based on Vitamin D and calcium; however, the dosage is unspecified, which can explain greater increase in BMD. In 2009, a 12-month study of 59 women with postmenopausal osteoporosis/osteopenia assessed the effect of resistance exercise on BMD in the femur and lumbar spine. The experimental group (57.5 ± 5.1 years) participated in a training program (divided in four stages), with the exercises performed in a closed kinematic chain. The exercises were divided into four stages; each stage lasted months and the number of repetitions progressively increased from 10 repetitions (in the first month) to 12 repetitions (the second month) to 15 (in the third month), with 1 min rest between sets and exercises. The control group (56.6 ± 4.6 years) did not participate in any exercise program during the 12 months and no group received drug treatment during the study. At the end of the 12 months, the experimental group recorded an increase (Δ% = 1.17) in BMD compared with the initial results in the lumbar spine (0.845 ± 0.09 vs. 0.855 ± 0.09, *p* = 0.22), whereas the control group recorded a statistically significant decrease in BMD in the lumbar spine (Δ% = −2.26, *p* = 0.019). The changes (post-test–pre-test) regarding the BMD (g/cm^2^) at the level of the lumbar spine were significantly different between the experimental group (0.010 ± 0.043) and the control group (−0.018 ± 0.039, *p* < 0.013) [24]. In the case of our implemented protocol, higher increases in lumbar spine BMD were observed compared with the study mentioned above (+1.17% vs. +1.82% in our case). We specify that the duration of the program used by Matos et al. was twice as long as ours, but the subjects did not receive Vitamin D treatment. Moreover, only weights of 1, 2, 3, and 4 kg were used, but the intensities used are not mentioned, meaning that we cannot make a comparison regarding this factor.

A one-year study conducted in 2013 separated the participants (postmenopausal women) into three groups: a control group (52 ± 3.4 years), a group participating in the resistance training (51.4 ± 2.7 years), and a group involved in a water exercise program (54.5 ± 3.3 years). BMD was assessed before the study and at its end. The resistance training program was performed three times a week, with a number of repetitions between 10–15 for each exercise. All the subjects underwent a hormone-based treatment, and at the end of the study, the group that followed the resistance training program recorded an increase in the lumbar spine L_2_–L_4_ of 16.40% (*p* < 0.05). Between the control group and the group participating in the resistance training program, the differences were significant in the lumbar spine (1283 ± 0.169 vs. 1070 ± 0.030, *p* < 0.001) [25]. Although increases in BMD are more evident in this study, the subjects in the group participating in strength training had been on a hormone-based treatment for 3 years (3.0 ± 1.7 years). The subjects of this study already registered a high BMD (because none of the subjects in the group who participated in the strength exercise program had osteopenia or osteoporosis; all of them had normal BMD).

Basat et al. conducted a 6-month study and compared the effects of training programs on BMD in postmenopausal women. In this study, one group participated in a resistance training program (55.9 ± 4.9 years; *n* = 11), one group participated in a program of exercises involving movements with high impact on bone-jumping (55.6 ± 2.9 years; *n* = 12), and a control group (56.2 ± 4.0 years; *n* = 12) did not exercise throughout the study; all groups received treatment with Vitamin D (800 IU) and calcium (1200 mg) daily. The training programs were conducted three times a week with 60 min session durations; after 15 min of warm-up, the subjects performed the isometric training program lasting 45–60 min, in which they performed one series of 10 repetitions for each exercise. At the end of the study, the group that participated in the resistance exercise program had an increase in BMD in the lumbar spine L_1_–L_4_ (Δ% = 1.3) compared with a decrease in BMD observed in the control group (Δ% = −2.5). Between the two groups, the differences were significant at the end of the study in the lumbar spine (*p* = 0.032) [26]. However, in this study, the subjects also underwent calcium-based treatment (and not just Vitamin D-based treatment, as in our case). The increase in lumbar spine BMD in our study is higher (+1.82%) compared to the increase (+1.3%) observed in the study conducted by Basat et al., although the duration of the intervention program was identical (6 months).

Exercise programs that do not involve weight training may also be effective in increasing BMD. Angin et al. showed significant differences after 6 months of practicing tai chi exercises compared with initial results on the BMD in the lumbar spine L_2_–L_4_ (0.673 ± 0.09 vs. 0.714 ± 0.09, *p* = 0.002), recording an increase of 0.04 ± 0.06 g/cm^2^, which is a 6.1% increase in BMD. However, in this study, the subjects were treated with bisphosphonates, which was also a criterion for including the subjects in the study [27].

## 5. Conclusions

This study shows the effect that our strength training protocol has on postmenopausal women with osteopenia or osteoporosis. Therefore, alternating loads of 70% of 1RM with loads of 50% of 1RM within the same set and performing the exercise protocol twice a week may lead to an increase in lumbar spine BMD for postmenopausal women with osteopenia/osteoporosis.

The method is also safe because the intensities do not exceed 70% of 1RM. Comparing the costs of drugs for treating osteopenia or osteoporosis with the costs involved in adopting such a training program indicates that this proposed method is an affordable strategy for the prevention of osteopenia or osteoporosis. In the future, we aim to evaluate the BMD and body composition of subjects (using bioimpedance devices) included in such a strength training protocol, as well as use specific dynamometers to measure muscle strength in order to assess to what extent muscle strength and muscle mass influence BMD for women with osteopenia or osteoporosis. We also want to evaluate the effect that this strength training protocol has on BMD in the hip, and its relationship to fat mass and lean mass for women with postmenopausal osteopenia/osteoporosis.

## Figures and Tables

**Figure 1 sensors-22-01904-f001:**
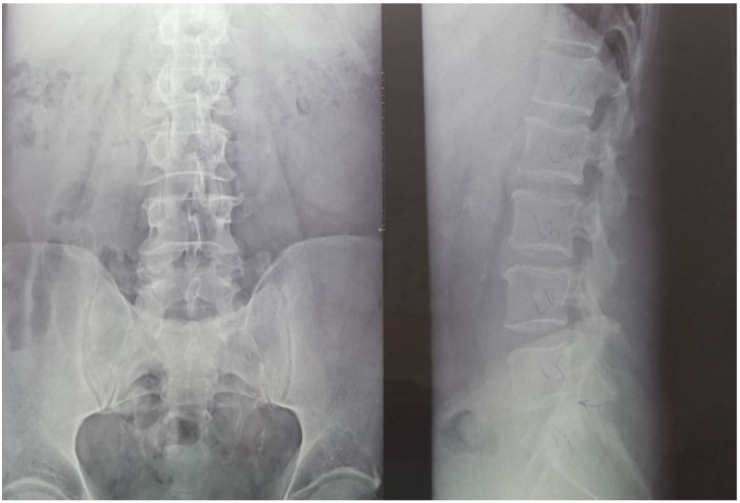
Radiological image of the subjects included in the study (X-ray of the lumbar spine). (**Left side**) an image in the frontal plane; (**right side**) an image in the sagittal plane (profile). Using radiography as an auxiliary method provides information on the height of the vertebrae, the intervertebral spaces, and on other possible associated problems that could be contraindications for participating in the training protocol.

**Figure 2 sensors-22-01904-f002:**
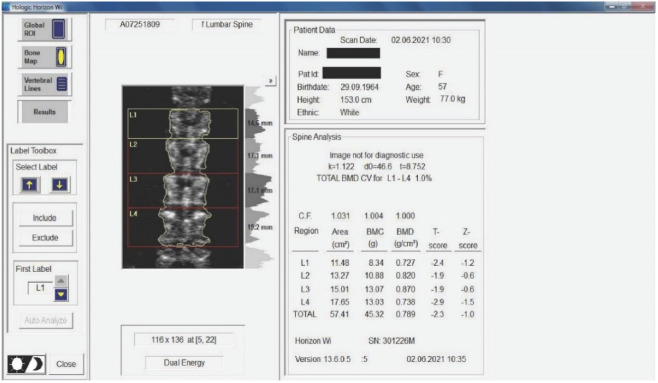
Measurement of BMD for the subjects included in the study. The results are displayed planimetrically, relating the BMC to the surface being evaluated (g/cm^2^). This also gives us a total T-score expressing whether the person has normal BMD (T-score between −1 and 0 or above), osteopenia (T-score between −1.1 and −2.4), or osteoporosis (T-score of −2.5 or less). The results in the image show us that the subject has osteopenia (but is extremely close to osteoporosis).

**Figure 3 sensors-22-01904-f003:**
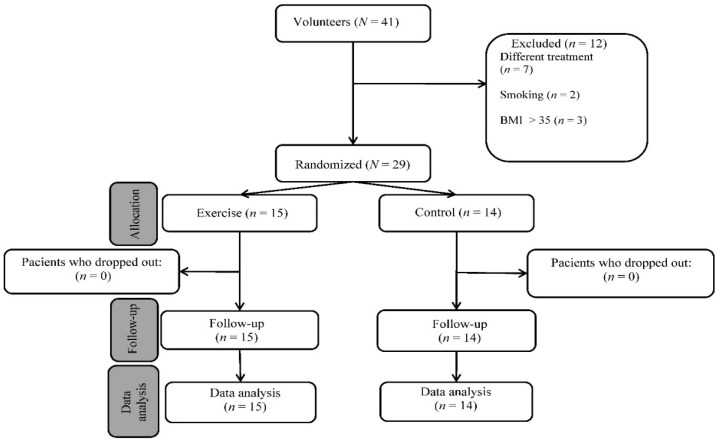
Flow diagram of study participants.

**Figure 4 sensors-22-01904-f004:**
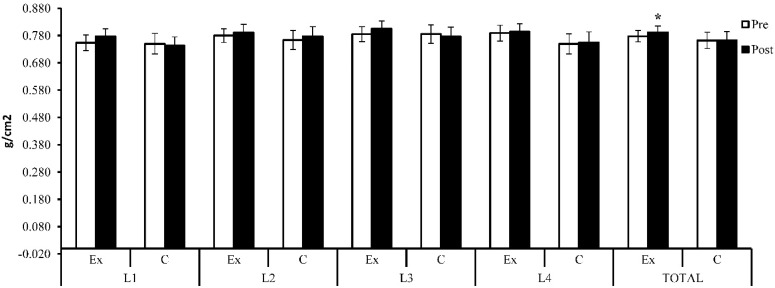
Lumbar spine BMD (g/cm^2^)—initial and baseline results; Ex = exercise group; C = control group. The symbol (*) indicates intra-group difference (*p* < 0.05).

**Table 1 sensors-22-01904-t001:** Baseline characteristics of the participants.

	Exercise (*n =* 15)	Control (*n =* 14)	*p* (Between Groups)
Age (years)	56.2 ± 3.2	56.8 ± 2.3	0.77
Weight (kg)	65.8 ± 7.4	63.2 ± 7.5	0.36
Height (cm)	161 ± 6.3	157.6 ± 4.7	0.12
BMI	25.4 ± 2.6	25.4 ± 2.1	0.99
BMD L_1_	0.754 ± 0.057	0.750 ± 0.072	0.87
BMD L_2_	0.780 ± 0.050	0.764 ± 0.067	0.46
BMD L_3_	0.785 ± 0.054	0.786 ± 0.064	0.97
BMD L_4_	0.789 ± 0.058	0.750 ± 0.070	0.11
BMD TOTAL	0.778 ± 0.042	0.762 ± 0.057	0.39

Note: results are represented as mean and standard deviation (±); BMI = body mass index; BMD = bone mineral density (g/cm^2^).

**Table 2 sensors-22-01904-t002:** Intra-group and inter-group comparison after 6 months.

	Exercise (*n =* 15)			Control (*n =* 14)			
	Pre	Post	*p* (Intra-Group)	Pre	Post	*p* (Intra-Group)	*p* (Inter-Groups)
Weight (kg)	65.8 ± 7.4	64.5 ± 7.4	0.008 *	63.2 ± 7.5	63.9 ± 7.6	0.065	0.85
Height (cm)	161 ± 6.3	160.8 ± 6.3	0.082	157.6 ± 4.7	157.5 ± 4.6	0.17	0.12
BMI	25.4 ± 2.6	24.9 ± 2.4	0.017 *	25.4 ± 2.1	25.7 ± 2.1	0.062	0.37
BMD L_1_	0.754 ± 0.057	0.777 ± 0.055	0.079	0.750 ± 0.072	0.743 ± 0.061	0.26	0.12
BMD L_2_	0.780 ± 0.050	0.791 ± 0.061	0.24	0.764 ± 0.067	0.777 ± 0.067	0.11	0.55
BMD L_3_	0.785 ± 0.054	0.805 ± 0.057	0.053	0.786 ± 0.064	0.776 ± 0.066	0.38	0.21
BMD L_4_	0.789 ± 0.058	0.794 ± 0.057	0.65	0.750 ± 0.070	0.755 ± 0.073	0.62	0.12
BMD TOTAL	0.778 ± 0.042	0.792 ± 0.046	0.018 *	0.762 ± 0.057	0.763 ± 0.059	0.85	0.15

* Statistical significance.

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
