# Peer review of "Effect of Strength Training Protocol on Bone Mineral Density for Postmenopausal Women with Osteopenia/Osteoporosis Assessed by Dual-Energy X-ray Absorptiometry (DEXA)"

_sensors, 2022, doi:10.3390/s22051904_

Round 1
Reviewer 1 Report
The aim of the research was “to use two imaging methods to diagnose osteopenia or osteoporosis, as well as to introduce a training program that has never been applied to women with postmenopausal osteopenia / osteoporosis to observe the impact it has on bone mineral density among these women”. The originality of the work should be clearly stated. Is there any novelty in the way you applied the two methods (DEXA and X-rays)? I have the following additional comments:
1) The title is: “Computer-Assisted Diagnosis to Assess Bone Mineral Density in Women with Osteopenia / Osteoporosis”. In lines 25-27, you mention “DEXA is a non-invasive computer- aided diagnosis (CAD) system”. Nevertheless, when reading the title, one may believe that the work will propose a CAD system, which is not the case so I would exclude this mention in the title and would include something related to the exercise protocol.
2) Abstract should contain the main numerical results and conclusion.
3) The objective should be at the end of the Introduction section and not in the Material and Methods section.
4) In Material and Methods section: please, give more details about what kind of quantitative information you took from DEXA (BMD? T-score? Z-score?)? You treated the vertebras L1-L4 individually? What kind of information you took from X-rays? Only visual information? No quantification of any kind? And how did you combine DEXA and X-rays information to bear the outcome? I did not see any results coming from X-rays of your subjects.
5) Equation 1: “XD = mean difference between the two sets of scores”. Which scores? Please define it.
6) Legend of Figs 1 and 2 should have more explanations to help interested readers who are not specialist.
7) In Figure 3, the first upper box has n = 29, then there is a box on the right presenting n = 12 excluded subjects, and then you randomize n = 29 patients. The original number of subjects should be 29+12 = 41, shouldn’t it?
8) Who prescribed the exercise protocol (duration, variety, pause between sets, and intensity)? And why this exercise protocol was chosen? This seems to be the originality of the work.
9) Results section. I recommend building a table with the values. The way results are presented it is very difficult to follow.
10) Discussion section:
a) Lines 172-218 talk about X-rays, since you do not have X-rays results from your subjects, you can diminish the details you present about this technique. One paragraph regarding the role and limitations of X-rays in osteoporosis identification is enough.
b) Lines 219-230: these lines are basically and explanation about DEXA and can be placed as the first paragraph of the Material and Methods section.
c) Lines 235-236: “The expression of the results is performed planimetrically, relating the bone mineral content to the surface we are evaluating (g/cm2).” It is a valuable information and should be put, for instance, in the legend of Figure 2.
d) The studies you mention (Lines, 267, 278, 297, 313) have to be compared to your results. And here there is an important choice to make: I see no originality in using DEXA itself so I believe your work would be more valuable if you compare the results of your proposed exercise protocol (in terms of BMD gain) to the other exercise protocols from the literature you cite.
11) Conclusion section: In this section you write the new information, based only on your original declared objectives.
a) Lines 347-349: “With the help of the DEXA investigation, important information can be obtained about bone mineral density or about its evolution to see if certain programs or treatments are effective”. This information was already known before you started your work. I would erase it.
b) Lines 315-352: “Thus, using the two methods together seems to be very effective in establishing the diagnosis of osteopenia or osteoporosis.”. I couldn’t find the results of X-rays for your subjects nor how you used both information to extract the results.
c) Lines 352-353: “This study also shows the effects that strength training has on women with osteopenia or osteoporosis in the postmenopausal period.”. This result seems to be the novelty of your work and, as I mentioned before, I would change the focus of your work to the exercise protocol results.
12) Please revise English language.
Author Response
Dear Reviewer,
Thank you for your valuable evaluation of our paper.
We used your recommendations in order to improve the quality of our article and also give us a fresh perspective to the topic that we approached.
There are the responses for each remark / suggestion.
The aim of the research was “to use two imaging methods to diagnose osteopenia or osteoporosis, as well as to introduce a training program that has never been applied to women with postmenopausal osteopenia / osteoporosis to observe the impact it has on bone mineral density among these women”. The originality of the work should be clearly stated. Is there any novelty in the way you applied the two methods (DEXA and X-rays)?
The novelty of our study is given by the two methods (DEXA and X-rays) and our strength training protocol.
I have the following additional comments:
1) The title is: “Computer-Assisted Diagnosis to Assess Bone Mineral Density in Women with Osteopenia / Osteoporosis”. In lines 25-27, you mention “DEXA is a non-invasive computer- aided diagnosis (CAD) system”. Nevertheless, when reading the title, one may believe that the work will propose a CAD system, which is not the case so I would exclude this mention in the title and would include something related to the exercise protocol.
We changed the title.
2) Abstract should contain the main numerical results and conclusion.
We added in abstract the main numerical results and conclusion.
3) The objective should be at the end of the Introduction section and not in the Material and Methods section.
We moved the objective of the study at the end of the Introduction.
4) In Material and Methods section: please, give more details about what kind of quantitative information you took from DEXA (BMD? T-score? Z-score?)? You treated the vertebras L1-L4 individually? What kind of information you took from X-rays? Only visual information? No quantification of any kind? And how did you combine DEXA and X-rays information to bear the outcome? I did not see any results coming from X-rays of your subjects.
We gave more details about quantitative information that we took from DEXA and X-rays.
5) Equation 1: “XD = mean difference between the two sets of scores”. Which scores? Please define it.
We defined the scores.
6) Legend of Figs 1 and 2 should have more explanations to help interested readers who are not specialist.
We added more explanation to each figure (1,2).
7) In Figure 3, the first upper box has n = 29, then there is a box on the right presenting n = 12 excluded subjects, and then you randomize n = 29 patients. The original number of subjects should be 29+12 = 41, shouldn’t it?
We corrected the original number of the subjects.
8) Who prescribed the exercise protocol (duration, variety, pause between sets, and intensity)? And why this exercise protocol was chosen? This seems to be the originality of the work.
We mentioned requested information.
9) Results section. I recommend building a table with the values. The way results are presented it is very difficult to follow.
We added 2 tables with results and statistical analysis.
10) Discussion section:
- a) Lines 172-218 talk about X-rays, since you do not have X-rays results from your subjects, you can diminish the details you present about this technique. One paragraph regarding the role and limitations of X-rays in osteoporosis identification is enough.
We diminished the X-rays details.
- b) Lines 219-230: these lines are basically and explanation about DEXA and can be placed as the first paragraph of the Material and Methods section.
We moved lines 219-230 at the beginning of the Materials and Methods section.
- c) Lines 235-236: “The expression of the results is performed planimetrically, relating the bone mineral content to the surface we are evaluating (g/cm2).” It is a valuable information and should be put, for instance, in the legend of Figure 2.
We moved lines 235-236 at the Figure 2 legend.
- d) The studies you mention (Lines, 267, 278, 297, 313) have to be compared to your results. And here there is an important choice to make: I see no originality in using DEXA itself so I believe your work would be more valuable if you compare the results of your proposed exercise protocol (in terms of BMD gain) to the other exercise protocols from the literature you cite.
We compared our results (exercise protocol) with other studies.
11) Conclusion section: In this section you write the new information, based only on your original declared objectives.
We added new information based on results of our exercise protocol.
- a) Lines 347-349: “With the help of the DEXA investigation, important information can be obtained about bone mineral density or about its evolution to see if certain programs or treatments are effective”. This information was already known before you started your work. I would erase it.
We erased lines 347-349.
- b) Lines 315-352: “Thus, using the two methods together seems to be very effective in establishing the diagnosis of osteopenia or osteoporosis.”. I couldn’t find the results of X-rays for your subjects nor how you used both information to extract the results.
We modified and added other information about DEXA and X-rays (lines 351-352) use in our study protocol and moved to Materials and Methods.
- c) Lines 352-353: “This study also shows the effects that strength training has on women with osteopenia or osteoporosis in the postmenopausal period.”. This result seems to be the novelty of your work and, as I mentioned before, I would change the focus of your work to the exercise protocol results.
We mentioned the benefits of our strength training protocol.
12) Please revise English language.
English language was revised.
Reviewer 2 Report
dear Authors,
The topic is interesting, but you need to improve the article in several parts.
Abstract: the methods and results must be more specifics.
Introduction: Needs to be more effective on justification to use radiography and DEXA (radiation) and a more extensive explaining about exercise. The aims must be reformulated. I think that you need to give more importance to the exercise.
Methods: ethical approve and informed consent are important to be in this topic.
Results: needs to empathise the exercise and the differences intergroup.
Conclusion: needs to be improved.
best regards
Author Response
Dear Reviewer,
Thank you for your valuable and professional remarks concerning our paper.
We used your recommendations in order to improve the quality of our article and also give us a fresh perspective to the topic that we approached.
The topic is interesting, but you need to improve the article in several parts.
Abstract: the methods and results must be more specifics.
We added the methods and specifics results of our study.
Introduction: Needs to be more effective on justification to use radiography and DEXA (radiation) and a more extensive explaining about exercise. The aims must be reformulated. I think that you need to give more importance to the exercise.
We reformulated the aim of the study.
We added new information concerning exercise protocol and it importance in Materials and Methods.
Methods: ethical approve and informed consent are important to be in this topic.
We added approve and informed consent.
Results: needs to empathise the exercise and the differences intergroup.
We added statistical data intra and inter-groups. We emphased our exercise protocol and compared with other studies.
Conclusion: needs to be improved.
We improved our conclusion and highlighted the benefits of our protocol.
Round 2
Reviewer 1 Report
The authors have presented a whole new approach to their work. The text is now focusing on the novelty. I believe it can now be accepted after some minor changes, as follows:
1) You may mention DEXA in the title and in the objective, as it is the method you used to generate your quantitative data.
2) There is no Table 2, please verify table numbering.
3) Please make a new revision of the English language.
Author Response
Dear Reviewer,
Thank you again for your remarks concerning our paper.
There are the responses for each remark / suggestion.
- You may mention DEXA in the title and in the objective, as it is the method you used to generate your quantitative data.
We mentioned DEXA in the title and in the objective.
- There is no Table 2, please verify table numbering.
That true, we check again and numbered the tables in our article.
3) Please make a new revision of the English language.
The English language was revised and corrected by a specialized service company – KG Support. The paper was revised and corrected by native English speakers editors.
I will ask for the Certificate if is needed.